# Discovery of a new class of reversible TEA domain transcription factor inhibitors with a novel binding mode

Lu Hu[1]*[†], Yang Sun[1†‡], Shun Liu[2†], Hannah Erb[1], Alka Singh[3], Junhao Mao[3], Xuelian Luo[2]*, Xu Wu[1]*

[1]Cutaneous Biology Research Center, Massachusetts General Hospital, Harvard Medical School, Charlestown, United States; [2]Departments of Pharmacology & Biophysics, University of Texas Southwestern Medical Center, Dallas, United States; [3]Department of Molecular, Cell and Cancer Biology, University of Massachusetts Medical School, Worcester, United States

*For correspondence:
LHU8@mgh.harvard.edu (LH);
xuelian.luo@utsouthwestern.
edu (XL);
xwu@cbrc2.mgh.harvard.edu
(XW)

[†]These authors contributed
equally to this work

Present address: [‡]Cancer
Institute, Xuzhou Medical
University, Xuzhou, China

Competing interest: See page
19

Reviewing Editor: Duojia Pan,
UT Southwestern Medical Center
and HHMI, United States

**Abstract** The TEA domain (TEAD) transcription factor forms a transcription co-activation complex with the key downstream effector of the Hippo pathway, YAP/TAZ. TEAD-YAP controls the expression of Hippo-responsive genes involved in cell proliferation, development, and tumorigenesis. Hyperactivation of TEAD-YAP activities is observed in many human cancers and is associated with cancer cell proliferation, survival, and immune evasion. Therefore, targeting the TEAD-YAP complex has emerged as an attractive therapeutic approach. We previously reported that the mammalian TEAD transcription factors (TEAD1–4) possess auto-palmitoylation activities and contain an evolutionarily conserved palmitate-binding pocket (PBP), which allows small-molecule modulation. Since then, several reversible and irreversible inhibitors have been reported by binding to PBP. Here, we report a new class of TEAD inhibitors with a novel binding mode. Representative analog TM2 shows potent inhibition of TEAD auto-palmitoylation both in vitro and in cells. Surprisingly, the co-crystal structure of the human TEAD2 YAP-binding domain (YBD) in complex with TM2 reveals that TM2 adopts an unexpected binding mode by occupying not only the hydrophobic PBP, but also a new side binding pocket formed by hydrophilic residues. RNA-seq analysis shows that TM2 potently and specifically suppresses TEAD-YAP transcriptional activities. Consistently, TM2 exhibits strong antiproliferation effects as a single agent or in combination with a MEK inhibitor in YAP-dependent cancer cells. These findings establish TM2 as a promising small-molecule inhibitor against TEAD-YAP activities and provide new insights for designing novel TEAD inhibitors with enhanced selectivity and potency.

## Editor's evaluation

In this article, Hu et al. describe the discovery and characterization of a new class of reversible TEAD inhibitors that binds to a novel side pocket adjacent to the palmitate-binding pocket. The newly identified highly tractable chemical matter and its novel binding mode provide an excellent starting point for the development of effective TEAD inhibitors.

## Introduction

Yes-associated protein (YAP) and transcriptional coactivator with PDZ-binding motif (TAZ) are the major downstream effectors of the evolutionarily conserved Hippo pathway that controls organ size and tissue homeostasis (*Pan, 2007*; *Yu et al., 2015*). Beyond their critical roles in development,

accumulating evidence shows that YAP/TAZ hyperactivation is frequently linked to tumorigenesis in a broad range of human cancers (*Harvey et al., 2013*; *Pan, 2010*; *Zanconato et al., 2016b*). Importantly, YAP/TAZ alone cannot interact with DNA; therefore, it requires the binding of transcriptional factors TEA/TEF-domain (TEAD1–4 in mammals and Scalloped in *Drosophila*) to regulate the expression of Hippo-responsive genes (*Wu et al., 2008*; *Zhao et al., 2008*). The transcriptional targets of the TEAD-YAP/TAZ complex are involved in cell proliferation, cell survival, immune evasion, and stemness (*Moroishi et al., 2015*). However, direct targeting YAP/TAZ with small molecules has been shown to be difficult. Therefore, pharmacological disruption of TEAD-YAP/TAZ has been considered as a promising avenue for cancer therapy (*Holden and Cunningham, 2018*; *Johnson and Halder, 2014*; *Pobbati and Hong, 2020*; *Zanconato et al., 2016a*).

One such strategy is to directly target TEAD-YAP interface with peptidomimetic inhibitors (*Jiao et al., 2014*; *Zhang et al., 2014*; *Zhou et al., 2015*). For instance, a peptide termed 'Super-TDU' was designed to block the TEAD-YAP interaction (*Jiao et al., 2014*). 'Super-TDU' mimics TDU domain of VGLL4, which competes with YAP/TAZ for TEAD binding, and has been shown to suppress gastric cancer growth. However, peptide-based inhibitors generally suffer from poor cell permeability and pharmacokinetic properties, limiting their therapeutic applications. Since TEAD-YAP binding interface is shallow and spanning a large surface area, it is particularly challenging to optimize small molecules for desired potency.

Previously, we and others discovered that TEAD auto-palmitoylation plays an important role in regulation of TEAD stability and TEAD-YAP binding, and loss of TEAD palmitoylation leads to inhibition of TEAD-YAP transcriptional activities (*Chan et al., 2016*; *Holden et al., 2020*). More importantly, structural and biochemical studies illustrated that the lipid chain of palmitate inserts into a highly conserved deep hydrophobic pocket (*Chan et al., 2016*; *Noland et al., 2016*), away from TEAD-YAP interface, which is suitable for small-molecule binding and suggests that lipid binding allosterically regulates TEAD-YAP activities.

Over the past years, targeting TEAD auto-palmitoylation has emerged as an attractive strategy for fighting cancers with aberrant YAP activation. To date, several companies and academic research groups have developed small-molecule inhibitors against TEAD-YAP activities. A nonsteroidal anti-inflammatory drug, flufenamic acid (FA), has been shown to bind to the lipid-binding pocket of TEAD (*Pobbati et al., 2015*). Although FA lacks potency to block TEAD function, it demonstrates that the lipid-binding pocket could indeed accommodate small-molecule binding. Ever since then, FA scaffold has been extensively explored by medicinal chemists to design TEAD inhibitors, including irreversible inhibitors TED-347 (*Bum-Erdene et al., 2019*), DC-TEADin02 (*Lu et al., 2019*), MYF-01-037 (*Kurppa et al., 2020*), K975 (*Kaneda et al., 2020*), as well as reversible inhibitor VT103 (*Tang et al., 2021*). In comparison, non-FA-based TEAD inhibitors are relatively limited, and only a few examples, such as compound 2, have been reported (*Holden et al., 2020*). Among the reported inhibitors, K975 and VT103 showed strong antiproliferation effects in vitro and antitumor effects in vivo. However, these inhibitors only have effects in limited cell lines, such as *NF2*-deficient mesothelioma cells. In addition, most of the reported TEAD inhibitors are irreversible inhibitors targeting the cysteine at the palmitoylation site, which might have undesired nonspecific reactivity toward other cysteines or other targets. To gain insights into the chemical diversity of reversible TEAD inhibitors and their utilities in cancer therapeutics, it is important to identify new chemical scaffolds to target TEADs.

We previously developed a non-FA-based reversible TEAD inhibitor, MGH-CP1 (*Li et al., 2020*), which inhibited transcriptional output of TEAD-YAP in vitro and in vivo. However, MGH-CP1 only showed sub-micromolar potency against TEAD palmitoylation in vitro and was used at low micromolar range in cellular assays. These limitations prompt us to develop new TEAD inhibitors with higher potency. In this study, we discovered a series of novel TEAD inhibitors featuring a common 4-benzoyl-piperazine-1-carboxamide scaffold. Among them, TM2 exhibits strong inhibition of TEAD2 and TEAD4 auto-palmitoylation in vitro with the IC50 values of 156 nM and 38 nM, respectively. In addition, palmitoylation of both exogenous Myc-TEAD1 and endogenous Pan-TEADs is also significantly diminished by TM2 in HEK293A cells, which further confirms its potency and mode of action in cellular context. The co-crystal structure of TEAD2 YBD in complex with TM2 uncovered a novel binding mode of the compound, which extended into a previously unknown hydrophilic side pocket adjacent to the PBP, and caused extensive side-chain rearrangements of the interacting residues. Further functional studies showed that TM2 significantly inhibits YAP-dependent liver organoid growth ex vivo

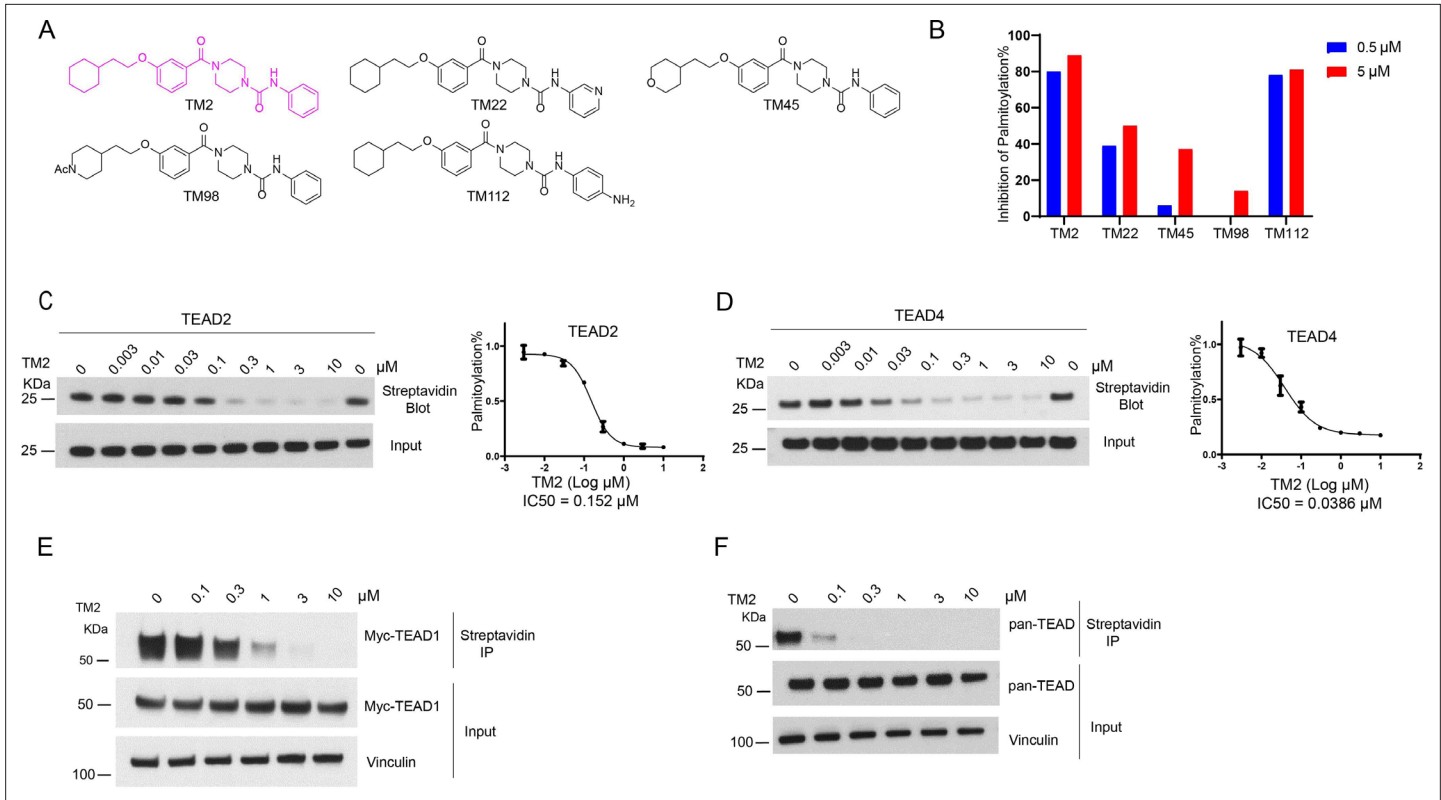

**Figure 1.** Identification of TM2 and analogs as novel TEA domain (TEAD) auto-palmitoylation inhibitors. (**A**) Representative chemical structures of a novel class of TEAD inhibitors with 4-(3-(2-cyclohexylethoxy)benzoyl)-piperazine-1-carboxamide moiety. TM2 structure is highlighted in magenta. (**B**) Inhibition of TEAD2 auto-palmitoylation with treatment of TM2 under 0.05 and 0.5 μM for 30 min, respectively. IC50 values for TM2 inhibition of TEAD2 (**C**) and TEAD4 (**D**) auto-palmitoylation were characterized by Western blot analysis (left) and quantified by ImageJ (right). The data was determined by independent replicates (n = 3) and shown as mean ± SEM. Palmitoylation of Myc-TEAD1 (**E**) and endogenous pan-TEAD (**F**) was analyzed by chemical reporter and streptavidin pulldown assay with treatment of TM2 at indicated concentrations for 24 h.

The online version of this article includes the following source data and figure supplement(s) for figure 1:

**Source data 1.** Source data for *Figure 1C–F*.

**Figure supplement 1.** Scheme for high-throughput screening of TEA domain (TEAD inhibitors).

**Figure supplement 2.** Comparison of TM2 and known inhibitors on inhibition of TEA domain (TEAD) palmitoylation.

**Figure supplement 2—source data 1.** Source data for *Figure 1—figure supplement 2*.

and inhibits proliferation of YAP-dependent cancer cells as a single agent or in combination with a MEK inhibitor. Overall, these studies broaden our understanding of the small-molecule-binding sites on TEADs.

## Results

### Identification of TM2 as a novel TEAD auto-palmitoylation inhibitor

To identify new chemotypes that could inhibit TEAD auto-palmitoylation, we screened a library containing about 30,000 non-proprietary medicinal chemistry compounds with three rounds of click-ELISA assay (*Lanyon-Hogg et al., 2015*) through the Astellas-MGH research collaboration by using the recombinant TEAD2 and TEAD4 YBD proteins. The inhibition of ZDHHC2 was used as a selectivity filter (*Figure 1—figure supplement 1*). We found several hits that share a common 4-(3-(2-cyclohexylethoxy)benzoyl)-piperazine-1-carboxamide moiety (data not shown, with micromolar potency in TEAD palmitoylation assays in vitro). The main variation is located at the *N*-substituent of the urea moiety with frequent incorporation of heteroarenes. Inspired by this structural convergence, we first designed a series of derivatives with variable substituents at the urea moiety, represented by TM2 and TM22 (*Figure 1A*). TEAD2 auto-palmitoylation in vitro assay was used to evaluate their

potency. Compared to heteroaryl group, phenyl substituent showed stronger inhibition on TEAD2 auto-palmitoylation (TM2 vs. TM22, *Figure 1B*). Inspired by these results, we explored the tolerance level by increasing hydrophilicity of TM2. As illustrated by TM45 and TM98, hydrophilic groups at the left cyclohexyl ring significantly decrease the activities, while the phenyl moiety at the right side of the urea moiety is well tolerated (TM112, *Figure 1B*). Overall, TM2 was identified as the most potent compound (*Figure 1B*) and selected for further biological evaluations.

TEAD family consists of four homologous members, TEAD1–4, which share highly conserved domain architectures (*Pobbati and Hong, 2013*). We found that TM2 inhibits TEAD2 palmitoylation with an $IC_{50}$ value of 156 nM (*Figure 1C*). Encouragingly, TM2 displays an even more potent effect on TEAD4 auto-palmitoylation with an $IC_{50}$ of 38 nM (*Figure 1D*). To study its effects on cellular TEAD palmitoylation, we overexpressed Myc-TEAD1 in HEK293A cells and treated with TM2 at different doses. As *Figure 1E* shows, TM2 dramatically suppresses Myc-TEAD1 palmitoylation in cells in a dose-dependent manner. Furthermore, treatment of TM2 also significantly inhibits endogenous TEAD1–4 palmitoylation using an antibody recognizing pan-TEADs (*Figure 1F*). Even at as low as 100 nM, Pan-TEAD palmitoylation was diminished. Collectively, these results suggested that TM2 is a potent and pan-inhibitor of palmitoylation of TEAD family proteins.

Furthermore, we compared its efficacy on inhibition of TEAD palmitoylation with known TEAD inhibitors, including an irreversible inhibitor K975, and reversible inhibitors VT103 and MGH-CP1. As shown in *Figure 1—figure supplement 2A*, TM2 is more potent than the other three TEAD inhibitors in TEAD2 auto-palmitoylation assay. Similar results were obtained when we compared their potencies on TEAD4 auto-palmitoylation (*Figure 1—figure supplement 2B*). Next, we evaluated their activities in HEK293A cells expressing full-length myc-TEAD1. TM2, K975, and VT103 markedly suppressed Myc-TEAD1 palmitoylation under indicated concentrations (*Figure 1—figure supplement 2C*). VT103 showed stronger activities on TEAD1 palmitoylation, consistent with the previous report that VT103 is a TEAD1-specific inhibitor (*Tang et al., 2021*). In addition, TM2 showed stronger activities in blocking endogenous pan-TEAD palmitoylation compared to other compounds (*Figure 1—figure supplement 2D*). Taken together, these data suggested that TM2 is a more potent and broader pan-inhibitor of TEADs than K975, VT103, and MGH-CP1.

## TM2 adopts a novel binding mode compared to other known TEAD inhibitors

To gain insights into the precise binding mode of TM2, we determined the co-crystal structure of TEAD2 YBD in complex with TM2 at 2.4 Å resolution (*Figure 2*, *Figure 2—figure supplement 1*, *Figure 2—source data 1*). Overall, TM2 binds to the same PBP in TEAD2 where palmitic acid (PLM) and other inhibitors target (*Figure 2A*). As shown in *Figure 2B*, the (2-cyclohexylethoxy)phenyl moiety of TM2 is surrounded by several hydrophobic residues, such as F233, L383, L390, F406, I408, Y426, and F428, enabling strong hydrophobic interactions, which is very similar to the interaction mode of TEAD with the fatty acyl chain of palmitic acid.

However, by superposing the TEAD2-TM2 (PDB 8CUH) with TEAD2-PLM structures (PDB 5HGU) (*Chan et al., 2016*), we observed a new feature of TM2 binding (*Figure 2B*). Unlike palmitic acid with its head group pointing toward residue C380, the urea moiety of TM2 exhibits a completely different orientation and sticks into a new side pocket, which has never been reported before to be involved in TEAD inhibitor binding and is only accessible by rearranging the side chains upon TM2 binding (*Figure 2B*, *Figure 2—figure supplement 1A*). TM2 binding drives significant conformational changes in the side chains of residues C343 and L374, which makes space for TM2 insertion (*Figure 2C*). Additionally, TM2 binding causes the side-chain movement in residues Q410 and Y333, which reduces the distance between the nitrogen atom of Q410 and the oxygen atom of Y333 from 4.9 Å to 2.7 Å to allow the formation of favorite electrostatic interaction (*Figure 2C*).

This binding model is highly consistent with our structure–activity relationship (SAR) results in *Figure 1A and B* that demonstrate that the left hydrophobic tail is repulsive to incorporate hydrophilicity, while the urea moiety is tolerated. The surface electrostatics of the TM2 binding pocket (*Figure 2—figure supplement 1A*) also illustrated that the (2-cyclohexylethoxy)phenyl moiety inserts into a nearly neutral environment, while the urea is buried in a pocket bearing electronegative properties. Furthermore, the electronegative carbonyl that links benzene and piperazine is spatially adjacent to electropositive electrostatics.

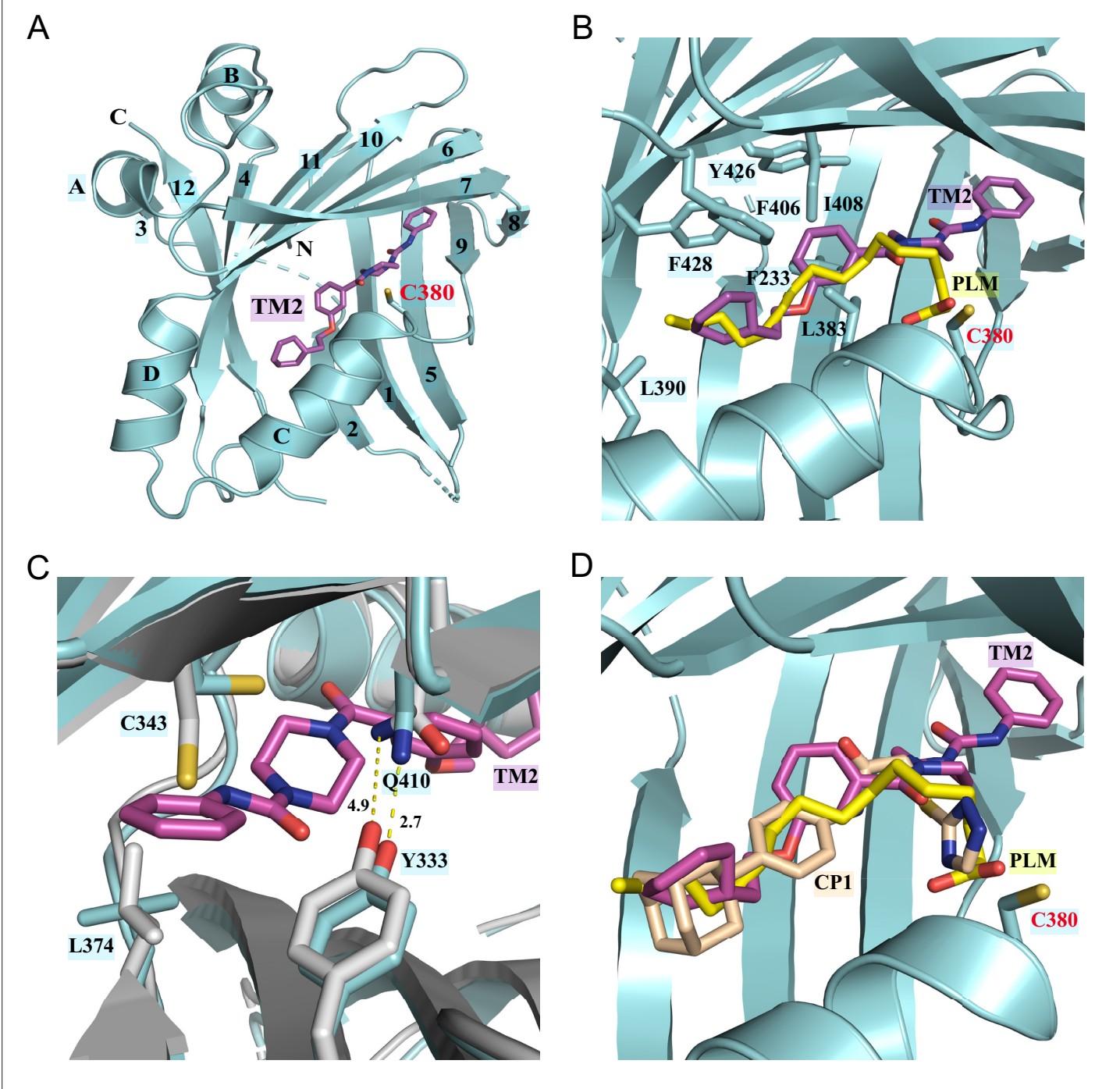

**Figure 2.** Co-crystal structure of TEAD2 complexed with TM2. (**A**) Ribbon diagram of the crystal structure of TEAD2-TM2 (PDB 8CUH). TM2 is shown as magenta sticks. (**B**) Close-up view of the TM2 binding site of TEAD2 (PDB 8CUH) with the superposition of the TEAD2-PLM structure (PDB 5HGU). Surrounding residues are shown as cyan sticks. Palmitic acid (PLM) is shown as yellow sticks. (**C**) Conformational changes in side chains of residues in the new pocket in the presence of TM2 binding. Indicated residues from TEAD2-TM2 and TEAD2-PLM are shown as cyan and gray sticks, respectively. Distances between atoms are shown with yellow dash lines and the unit is angstrom. (**D**) Structural superposition of TEAD2-TM2 (PDB 8CUH), TEAD2-PLM (PDB 5HGU), and TEAD2-CP1(PDB 6CDY). TEAD2 is shown as cyan ribbon. TM2, PLM, and CP1 (MGH-CP1) are shown as sticks and colored in magenta, yellow, and wheat, respectively.

The online version of this article includes the following source data and figure supplement(s) for figure 2:

**Source data 1.** Data collection and structure refinement statistics.

**Figure supplement 1.** Structural superposition suggests that TM2 adopts a novel binding model.

**Figure supplement 2.** The side pocket is conserved among TEA domain (TEAD) family protein.

We then assessed whether this unexpected binding model is unique to TM2 compared to other known TEAD inhibitors. Given that reported TEAD inhibitors are co-crystallized with different members of TEAD family of proteins, we aligned crystal structures of TEAD1–4 (*Figure 2—figure supplement 2A*). It showed that all eight interacting residues in the new side pocket of TEAD2 were highly conserved among all the TEAD family members. Besides, the protein sequence alignment also demonstrated that the key residues within the newly identified binding site were conserved, even in TEADs from other species (*Figure 2—figure supplement 2B*). Although there are some variants, for example, Cys343 is a Val in TEAD3 and Q410 is Leu in TEAD1, the binding affinity should not be affected. Under this premise, the co-crystal structures of TEAD YBD in complex with PLM (PDB 5HGU), TM2 (PDB 8CUH), and other known TEAD inhibitors, including MGH-CP1 (PDB 6CDY) (*Li et al., 2020*), K975 (PDB 7CMM) (*Kaneda et al., 2020*), and VT105 (PDB 7CNL) (*Tang et al., 2021*), were superposed (*Figure 2D*, *Figure 2—figure supplement 1B and C*). Consistent with previously reported results, MGH-CP1, VT105, or K975 adopts almost the same binding mode as PLM and fits very well with the PBP. However, the scenario depicted by TM2 is quite different, which provides new insights into the structural adaptability for the development of TEAD inhibitors. Considering relatively higher hydrophilicity in the new side pocket, there will be much more space to balance the lipophilicity of TEAD inhibitors and improve drug-like properties, such as solubility and metabolism (*Waring, 2010*).

## TM2 inhibits TEAD-YAP association and TEAD-YAP transcriptional activity

TEAD auto-palmitoylation plays an important role in the regulation of TEAD-YAP interaction. To confirm whether TM2 functions through blockade of TEAD-YAP binding, we tested TM2 in malignant pleural mesothelioma (MPM) cell line NCI-H226 cells, which is deficient with NF2 and highly dependent on TEAD-YAP activities (*Kaneda et al., 2020*; *Tang et al., 2021*). YAP co-immunoprecipitation (IP) experiments indicated that TM2 dramatically blocked the association of YAP with endogenous TEAD1 as well as pan-TEAD in a dose-dependent manner (*Figure 3A*). Next, we evaluated the effects of TM2 in the expressions of TEAD-YAP target genes, represented by *CTGF*, *Cyr61*, and *ANKDR1*. After treatment of TM2, the expression levels of *CTGF* and *ANKDR1* were significantly suppressed at both 24 and 48 hr, while *Cyr61* showed strong response at 48 hr (*Figure 3B*, *Figure 3—figure supplement 1*).

In order to systemically evaluate the effect of TM2 on YAP/TAZ-TEAD transcriptional activation, we performed RNA-seq analysis (*Figure 3C*). YAP/TAZ-dependent NCI-H226 cells were treated with or without TM2. We performed principal component analysis (PCA), a mathematical algorithm reducing the dimensionality of the data while retaining most of the variation in the data sets. The samples were plotted and indicated that TM2 treatment substantially altered the gene sets at PC1 in NCI-H226 cells (*Figure 3D*). Gene set enrichment analysis (GSEA) was performed to analyze the transcriptional signature gene sets from Molecular Signature Database. It showed that YAP signature was among the top 3 enriched signatures according to the Normalized Enrichment Score (NES) (*Figure 3E*). To further validate the effects of TM2 on YAP/TAZ signaling, the Cordernonsi_YAP_conserved_Signature and YAP_TAZ-TEAD Direct Target Genes were determined (*Zanconato et al., 2015*). Consistently, YAP/TAZ signature was significantly enriched in downregulation phenotype in both gene sets (*Figure 3F*). We then compared the effects of TM2 on genes modulation with that of two advanced TEAD inhibitors, irreversible inhibitor K975 and reversible inhibitor VT103, which showed strong antitumor effects in NCI-H226 xenograft tumor (*Kaneda et al., 2020*; *Tang et al., 2021*). In global analyses of genes transcriptional alteration in NCI-H226 cells treated with TM2, K975, and VT103, we found that these three compounds adopt a similar pattern on regulation of global gene expressions (*Figure 3—figure supplement 2*). To further assess their modulation of Hippo pathway transcriptional outputs, we compared the transcriptional responses of YAP/TAZ-TEAD target genes in NCI-H266 upon the treatment of these compounds. TM2 showed comparable effects with K975 on target genes regulation, visualized by heatmap analysis (*Figure 3G*), and both compounds showed stronger modulation of YAP-target genes than VT103. Furthermore, we carried out GSEA of TEAD-YAP/TAZ target genes with TM2, K975, and VT103 treatment, and the NES are −2.64,–2.56, and –2.31, respectively, for the three compounds (*Figure 3—figure supplement 3*), suggesting that TM2 likely showed better TEAD-YAP/TAZ target genes enrichment. Taken together, we identified TM2 as a potent TEAD inhibitor, with strong modulation of TEAD-YAP/TAZ transcriptional activities.

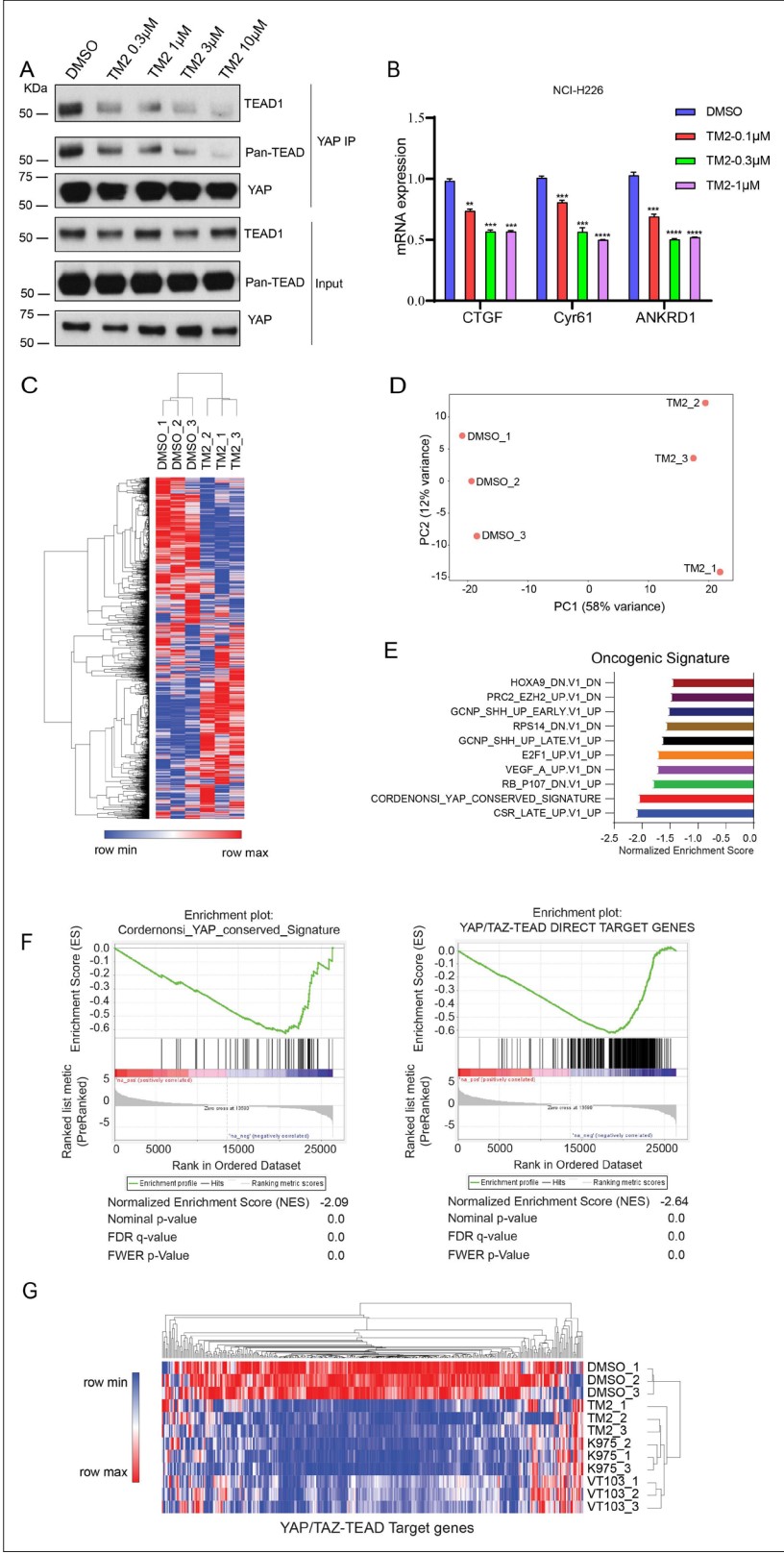

**Figure 3.** TM2 suppressed transcriptional outputs of Hippo pathway in cancer cells. (**A**) NCI-H226 cells were treated with TM2 at indicated concentrations for 24 hr. The interactions of YAP and Pan-TEAD as well as TEAD1 were observed with YAP co-IP. (**B**) Representative target genes of Hippo pathway in NCI-H226 cells were measured with treatment of TM2 at indicated concentrations for 48 hr. The data was determined by independent replicates (n

*Figure 3 continued on next page*

*Figure 3 continued*

= 3) and shown as mean ± SEM. Significance was determined by two-tailed *t*-test. **p<0.01, ***p<0.001, ****p<0.0001. (**C**) Heatmap analysis of biological replicates (n = 3) of global genes transcriptional alteration in NCI-H226 treated with vehicle control or TM2. Each column in the heatmap is an individual sample. (**D**) Principal component analysis (PCA) biplot with genes plotted in two dimensions using their projections onto the first two principal components, and six samples (Control: three samples; TM2: three samples) plotted using their weights for the components. (**E**) Gene set enrichment analysis of NCI-H226 cells treated with TM2 using oncogenic signature gene sets from Molecular Signatures Database. (**F**) Gene set enrichment plot of Cordernonsi_YAP_conserved_Signature (left panel) and YAP_TAZ-TEAD Direct Target Genes (right panel) with NCI-H226 cells treated with TM2. (**G**) Heatmap analysis of YAP/TAZ-TEAD direct target genes transcriptional alteration in NCI-H226 cells treated with TM2 (1 µM), K975 (1 µM), and VT103 (1 µM) for 24 h. Each column in the heatmap is an individual sample.

The online version of this article includes the following source data and figure supplement(s) for figure 3:

**Source data 1.** Source data for *Figure 3A*.

**Figure supplement 1.** Target gene expression in NCI-H226 with TM2 treatment for 24 hr.

**Figure supplement 2.** Heatmap analysis of global genes transcriptional alteration in NCI-H226 cells treated with TM2, K975, and VT103 for 24 hr.

**Figure supplement 3.** Gene set enrichment plot of YAP_TAZ-TEAD Direct Target Genes in NCI-H226 cells treated with K975 (left) and VT103 (right).

## TM2 inhibits YAP-dependent organoids' growth and cancer cell proliferation

YAP activity has been shown to be critical for the growth of liver organoid (*Planas-Paz et al., 2019*). Therefore, we used mouse hepatic progenitor ex vivo organoids to further investigate the effects of TM2 in a physiologically relevant model. As shown in *Figure 4A*, TM2 impaired the sustainability of organoids' growth in a dose-dependent manner, with more than 85% of disruption at 40 nM. Consistently, Ki67-positive cells for organoids maintenance in 3D culture were significantly diminished upon TM2 treatment (*Figure 4B*, *Figure 4—figure supplement 1*). To study the effects of TM2 on organoid growth, we measured the size of hepatic organoids. Our results showed that organoid size was significantly reduced with TM2 treatment (*Figure 4—figure supplement 2A*), indicating that TM2 inhibited organoids' growth. These data are consistent with previous observation that YAP activation leads to overgrowth of liver organoids (*Yimlamai et al., 2014*). Recent evidence demonstrated that activation of Hippo signaling is required to maintain differentiation states of hepatocyte (*Lee et al., 2016*; *Yimlamai et al., 2014*). Therefore, we then investigated whether inhibition of TEAD-YAP activity will affect the fate of organoids. We detected expression levels of mature hepatocyte markers, such as *Alb*, *Fxr*, and *cyp3a*. TM2 only showed minor effects on *cyp3a* expression, but significantly induced the expressions of *Alb* and *Fxr* (*Figure 4—figure supplement 2B*), suggesting that TM2 might induce hepatic differentiation of the organoid. Collectively, these data suggested that TM2-mediated TEAD inhibition is functional in a physiologically relevant YAP-dependent model.

Pleural mesothelioma (MPM) is a type of aggressive tumor, associated with exposure to asbestos fibers (*Rossini et al., 2018*). Despite several standard therapies, such as surgery, radiotherapy, chemotherapy, and immunotherapies, MPM patients still suffer poor prognosis with a median survival of only 8–14 months (*Nicolini et al., 2019*). *NF2* and *LATS2*, the upstream components of Hippo pathway, are frequently observed to be inactivated in malignant mesothelioma (MM), leading YAP activation in more than 70% of analyzed primary MM tissues (*Murakami et al., 2011*; *Sekido, 2018*). Therefore, MM would be a good model to study the therapeutic effects of TM2 on Hippo signaling-defective cancers. Encouraged by the strong inhibition of TEAD-YAP transcriptional activities in NCI-H226 cells, we first evaluated antiproliferative activities of TM2 in this cell line. As shown in *Figure 4C*, NCI-H226 cells exhibited striking vulnerability to TM2 treatment with an IC$_{50}$ value of 26 nM, consistent with its potency in blocking TEAD palmitoylation in vitro and in cells. Other derivatives, including TM22, TM45, TM98, and TM112, are less potent as TM2, which correlated well with their in vitro activities (*Figure 4C*). In addition, we also studied the effects of TM2 in two other MPM cell lines, MSTO-211H and NCI-H2052, which harbors *Lats1/2* deletion/mutations and *NF2* deficiency, respectively (*Kaneda et al., 2020*; *Lin et al., 2017*; *Miyanaga et al., 2015*). Consistently, TM2 also significantly inhibits cell proliferation of MSTO-211H and NCI-H2052 cells (*Figure 4D*) with IC$_{50}$ values of 94 nM and 157 nM,

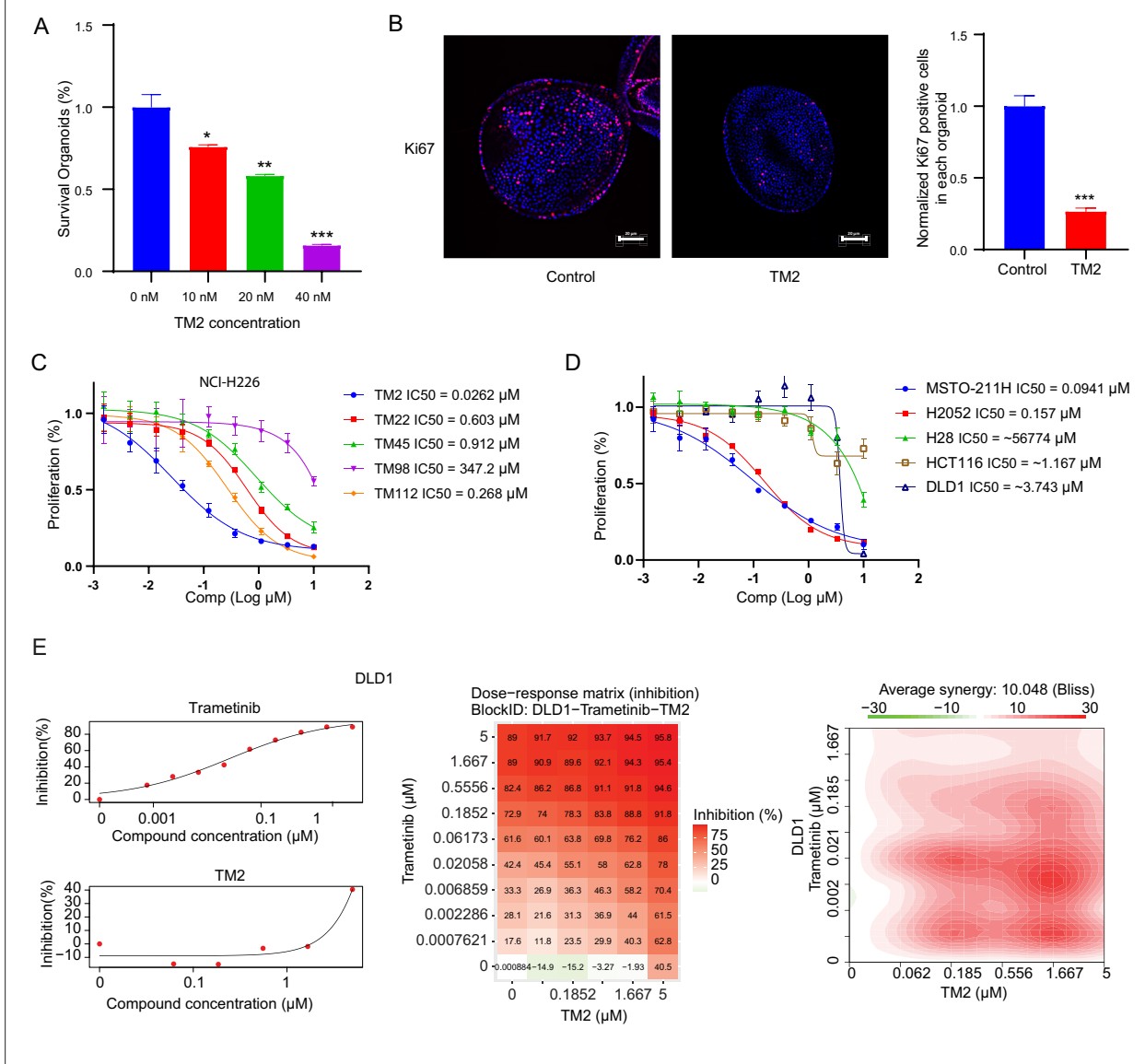

**Figure 4.** TM2 showed inhibition on YAP-dependent proliferation. (**A**) Percentages of survival organoids with treatment of control or TM2 at indicated concentrations. The data was determined by independent replicates (n = 3) and shown as mean ± SEM. Significance was determined by two-tailed *t*-test. \*p<0.05, \*\*p<0.01, \*\*\*p<0.001 (**B**) Representative Immunofluorescent staining of Ki67 in organoids treated with control or TM2 (40 nM). Pink, Ki-67; blue, nuclear DNA (DAPI). Bar, 20 μm. Bar graphs showing the normalized percentage of positive cells in each organoid. Data are determined by independent replicates (n = 3) and represented as mean ± SEM. Significance was determined by two-tailed *t*-test. Significance was determined by two-tailed *t*-test. \*\*\*p<0.001. (**C**) Cell inhibition in NCI-H226 cells with treatment of compounds at indicated concentrations for 6 days. The data was determined by independent replicates (n = 3) and shown as mean ± SEM. (**D**) Cell inhibition in MSTO-211H, H2052, H28, HCT116, and DLD1 cells with treatment of TM2 at indicated concentrations for 5, 7, 6, 5, or 5 days, respectively. The data was determined by independent replicates (n = 3) and shown as mean ± SEM. (**E**) Drug combination experiments using TM2 and MEK inhibitor trametinib in DLD1. Left panels show the dose curves of the single agents. Middle panel shows the heatmap with color-coding as percentage of cell viability normalized to untreated controls under combination. Right panel shows the heatmap of Bliss score for TM2 and trametinib combination. Bliss score of >10 indicates synergistic effects.

The online version of this article includes the following figure supplement(s) for figure 4:

**Figure supplement 1.** Representative bright-field images of organoids treated with control or TM2 (40 nM).

**Figure supplement 2.** The effects of TM2 on organoid proliferation and differentiation.

**Figure supplement 3.** Comparison of the antiproliferative effects of TM2 and K975 in mesothelioma with YAP dependency.

**Figure supplement 4.** Drug combination experiments using TM2 and MEK inhibitor trametinib in HCT116 cells.

respectively. In comparison, TM2 shows no significant inhibition in the Hippo WT mesothelioma cells, NCI-H28 with IC$_{50}$ >5 µM (*Tanaka et al., 2015*; *Figure 4D*), suggesting that TM2 is specific to YAP-activated cancer cells. We then compared the antiproliferation effects of TM2 with K975. In NCI-H226 cells, TM2 showed potent and comparable activity with K975 (*Figure 4—figure supplement 3A*). Consistent with the observation in NCI-H226, the covalent inhibitor K975 is also as potent as TM2 in MSTO-211H cells (*Figure 4—figure supplement 3B*), while VT103 only showed partial inhibition with a flat curve (data not shown). Overall, TM2 and K975 displayed comparable inhibitory effects on cell proliferation in YAP-dependent mesothelioma cells. Given that irreversible inhibitor might harbors intrinsic off-target disadvantages than a reversible inhibitor, a more potent reversible, such as TM2, might serve as a more promising therapeutic agent for TEAD-YAP-dependent cancers.

Currently, TEAD inhibitors mainly show promising therapeutic potentials in mesothelioma, with limited activities in other YAP-dependent cancer cells. Given that deregulated Hippo signaling is implicated in many human cancers (*Harvey et al., 2013*), it is important to test the efficacy of TEAD inhibitors in cancers beyond mesothelioma, which will deepen our understanding of therapeutic spectrum of blocking TEAD-YAP activities. Therefore, we evaluated TM2 in colorectal cancer (CRC) as Hippo pathway has been shown to regulate the progression of CRC (*Della Chiara et al., 2021*; *Jin et al., 2021*; *Pan et al., 2018*). However, TM2 did not exhibit strong inhibition on cell proliferation of two CRC cell lines (*Figure 4D*), HCT116 and DLD1. These results suggested that suppression of Hippo transcriptional activities in CRC alone might not be sufficient to inhibit cell growth, as observed in mesothelioma. Indeed, YAP are found to be capable of rescuing cell viability in HCT116 with loss of function of KRAS, implying that KRAS signaling might also account for lack of potency of TM2 in CRC. Hence, we performed a drug combination matrix analysis across five doses of TM2 and nine doses of MEK inhibitor trametinib in HCT116 and DLD1, respectively. Encouragingly, we observed strong inhibitory effects and substantial synergy in both of the two cell lines (*Figure 4E*, *Figure 4—figure supplement 4*), suggesting that combining TEAD inhibitors with other therapies might be a good strategy to broaden their therapeutic applications in near future. Together, our data highlights that TM2 might have appealing potential to antagonize carcinogenesis driven by aberrant YAP activities.

## Discussion

In this study, we discovered a new class of reversible pan-TEAD inhibitors. The most potent compound, TM2, significantly diminished TEAD2/4 auto-palmitoylation in nanomolar ranges. Co-crystal structure analysis of TM2 in complex with TEAD2 YBD discovered a novel binding mode. It showed that the much more hydrophobic part of TM2 featured a cyclohexyl ring, exquisitely fitting the palmitoylation pocket, which is the most well-known structure feature for targeting TEAD (*Dey et al., 2020*). Surprisingly, the structure demonstrates that the urea moiety of TM2 does not overlap with PLM, but sticks into a new and unique side pocket. This binding site is not occupied by all other known TEAD inhibitors, such as MGH-CP1, VT105, and K975. This side pocket is not fully available in the palmitate-bound TEAD2 structure, but is formed with significant side-chain rearrangement upon TM2 binding. Moreover, this side pocket is endowed with higher hydrophilicity than the lipid-binding pocket, providing potentials for enhancing drug-like properties. The novel binding model expands structural diversity of the TEAD-binding pocket and will boost the discovery of more novel chemotypes, contributing to the development of therapeutics targeting TEAD-YAP.

Blocking TEAD auto-palmitoylation by TM2 disrupted TEAD-YAP association. Consistently, we observed significant suppression of downstream Hippo transcription program with treatment of TM2. RNA-seq analysis further confirmed that TM2 specifically inhibits YAP transcriptional signatures. YAP/TAZ is constitutively active in many human malignancies and shown to be essential for many cancer hallmarks (*Zanconato et al., 2016b*). Therefore, targeting YAP/TAZ activities has been considered as an attractive strategy for cancer therapy. In human MPM, a type of tumor that is highly associated with YAP activation, TM2 showed striking antiproliferation efficacy as a single agent, which is consistent with the fact that therapeutic effects of TEAD inhibitors are mainly limited to mesothelioma models (*Kaneda et al., 2020*; *Tang et al., 2021*). In colorectal cancer HCT116 and DLD1, single treatment of TM2 was insufficient to inhibit their growth, although they are also reported to be dependent on YAP activities. This might be interpreted by the activation of other oncogenic signaling pathways in these cancers, including Ras-MAPK activations. Indeed, YAP has been shown to converge with KRAS and can rescue cell viability induced by KRAS suppression (*Shao et al., 2014*), suggesting that inhibiting

YAP activities might be also rescued by other oncogenes. Consistently, significant synergy effects were observed when combining TM2 with a MEK inhibitor. These encouraging results suggested that rationalized combination of TEAD inhibitors with other inhibitors could significantly expand the utilities. In summary, our study disclosed TM2 as a promising new starting point for developing novel antitumor therapeutics against TEAD-YAP activities.

## Materials and methods
### Inhibition of TEAD2 and TEAD4 auto-palmitoylation in vitro
Recombinant 6xHis-TEAD protein was treated with compounds under indicated concentrations in 50 mM MES buffer (pH 6.4) for 30 min. After incubation with 1 µM of alkyne palmitoyl-CoA (15968, Cayman) for 1 hr, 50 µL of sample mixture was treated with 5 µL of freshly prepared 'click' mixture containing 100 uM TBTA (678937, Sigma-Aldrich), 1 mM TCEP (C4706, Sigma-Aldrich), 1 mM $CuSO_4$ (496130, Sigma-Aldrich), 100 uM Biotin-Azide (1167-5, Click Chemistry Tools), and incubated for another 1 hr. The samples were then added 11 µL of 6xSDS loading buffer (BP-111R, Boston BioProducts) and denatured at 95°C for 5 min. SDS-PAGE was used to analyze the samples. Palmitoylation signal was detected by streptavidin-HRP antibody (1:3000, S911, Invitrogen). The total protein level was detected by primary anti-His-tag antibody (1:10,000, MA1-21315, Invitrogen) and secondary anti-mouse antibodies (1:5000, 7076S, Cell Signaling). The band intensities were quantified with ImageJ. The inhibition of auto-palmitoylation by compounds was normalized to DMSO. The IC50 curves were plotted with GraphPad Prism6.

### Cell lines
Human NCI-H226, HEK293A, MSTO-211H, H2052, H28, HCT116, and DLD1 cells were obtained from ATCC (Manassas, VA). ATCC human cell authentication assay identifies short tandem repeat (STR) markers. None of the cells is listed in ICLAC as misidentified cell line. HEK293A, HCT116, and DLD1 cells were cultured in Dulbecco's Modified Eagle Medium (DMEM) (Life Technologies) supplemented with 10% (v/v) fetal bovine serum (FBS) (Thermo/Hyclone, Waltham, MA), 100 units/mL penicillin, and 100 µg/mL streptomycin (Life Technologies) at 37°C with 5% $CO_2$. NCI-H226, MSTO-211H, H2052, and H28 cells were cultured in RPMI 1640 medium (Life Technologies) supplemented with 10% (v/v) FBS (Thermo/Hyclone, Waltham, MA), 100 units/mL penicillin, 100 µg/mL streptomycin (Life Technologies), 2.5 g/L glucose, and 1 mM sodium pyruvate at 37°C with 5% $CO_2$. All cells are free of mycoplasma contamination.

### Transfection
HEK293A cells were seeded in 6 cm dishes overnight and transfected with plasmids using PEI reagent (1 µg/µL). Briefly, PRK5-Myc-TEAD1 (33109, Addgene) and PEI were diluted in serum-free DMEM medium in two tubes (DNA:PEI ratio = 1:2). After standing still for 5 min, mix them well and stay for another 20 min. The mixture was then added to dishes directly.

### Inhibition of TEAD palmitoylation in HEK293A cells
HEK293A cells with or without TEAD overexpression were pretreated with DMSO or compounds in medium with 10% dialyzed fetal bovine serum (DFBS) for 8 hr and labeled by alkynyl palmitic acid (1165, Click Chemistry Tools) for another 16 hr. The cells were then washed and harvested by cold DPBS (14190250, Life Technologies). The cell pellets were isolated by centrifugation (500 × $g$, 10 min) and lysed by TEA lysis buffer (50 mM TEA-HCl, pH 7.4, 150 mM NaCl, 1% Triton X-100, 0.2% SDS, 1× protease inhibitor-EDTA free cocktail [05892791001, Roche], phosphatase inhibitor cocktail [P0044, Sigma-Aldrich]) on ice for 30 min. The protein concentration is determined using Bio-Rad assay and adjusted to 1 mg/mL. Then, 100 µL of protein sample mixture was treated with 10 µL of freshly prepared 'click' mixture containing 1 mM TBTA, 10 mM TCEP, 10 mM $CuSO_4$, 1 mM TBTA Biotin-Azide, and incubated for 1 hr at room temperature. The proteins were precipitated by chloroform/methanol/$H_2O$ mixture and redissolved with 2% SDS in 0.1% PBST. The solution was diluted with 0.1% PBST and incubated with prewashed streptavidin agarose beads (69203-3, EMD Millipore). After rotation at room temperature for 2 hr, the beads were then pelleted by centrifugation (500 × $g$, 3 min) and washed with 0.2% SDS in PBS (3 × 1 mL). The bound proteins were eluted with a buffer containing

10 mM EDTA pH 8.2 and 95% formamide and analyzed with SDS-PAGE. Anti-Myc (1:1000, 2278S, Cell Signaling) or anti-pan-TEAD (1:1000, 13295, Cell Signaling) antibody was used to detect Myc-TEAD1 or pan-TEAD, respectively. Secondary antibody was anti-rabbit (1:5000, 7074S, Cell Signaling).

## Protein purification, crystallization, and structure determination

The recombinant human TEAD2 (residues 217–447, TEAD2 217–447) protein was purified and crystallized as described previously (*Li et al., 2020*). Single crystals were soaked overnight at 20°C with 5 mM TM2, 5% DMSO in reservoir solution supplemented with 25% glycerol and flashed-cooled in liquid nitrogen. Diffraction data was collected at beamline 19-ID (SBC-XSD) at the Advanced Photon Source (Argonne National Laboratory) and processed with HKL3000 program (*Otwinowski and Minor, 1997*). Best crystals diffracted 2.40 Å and exhibited the symmetry of space group C2 with cell dimensions of a = 124.1 Å, b = 62.3 Å, c = 79.9 Å, and $\beta$ = 117.7°. Using TEAD2 structure (PDB ID: 3L15) as searching model, initial density map and model were generated by molecular replacement with Phaser in PHENIX (*Adams et al., 2010*). There are two TEAD2 molecules in the asymmetric unit. One TM2 molecule was built in the cavity of each TEAD2 molecule, and the remaining residues were manually built in COOT39 and refined in PHENIX. The final model (Rwork = 0.184, Rfree = 0.235) contains 400 residues, 30 water molecules, and 2 TM2 molecules. Statistics for data collection and structure refinement are summarized in *Figure 2—source data 1*. The structure has been validated by wwPDB (*Berman et al., 2003*). Atomic coordinates and structure factors have been deposited to the Protein Data Bank under code 8CUH. Structural analysis and generation of graphics were carried out in PyMOL.

## Co-IP assay

NCI-H226 cells were treated with DMSO or TM2 for 24 hr. The cells were then washed and harvested by cold DPBS. The cell pellets were isolated by centrifugation (500 × *g*, 10 min) and lysed by lysis buffer (50 mM Tris–HCl pH 7.5, 10% glycerol, 1% NP-40, 300 mM NaCl, 150 mM KCl, 5 mM EDTA, phosphatase inhibitor cocktail, and complete EDTA-free protease inhibitors cocktail) on ice. After dilution with 50 mM Tris–HCl pH 7.5, 10% glycerol, 1% NP-40, and 5 mM EDTA, the protein samples were incubated with mouse anti-YAP antibody (sc-101199, Santa Cruz) overnight at 4°C and immunoprecipitated with prewashed protein A/G beads (P5030-1, UBPBio) for another 4 hr at 4°C. The bound proteins were washed with 0.1% PBST for three times and eluted with 1× SDS loading buffer and analyzed with SDS-PAGE. Anti-TEAD1 (1:1000, 12292S, Cell Signaling), anti-pan-TEAD (1:1000, 13295, Cell Signaling), or anti-YAP (1:1000, 14074S, Cell Signaling) antibody were used to detect TEAD1, pan-TEAD, or YAP, respectively. Secondary antibody was anti-rabbit (1:5000, 7074S, Cell Signaling).

## Quantitative RT-PCR

NCI-H226 cells or organoids were treated with DMSO or TM2 under indicated conditions and used to extract RNA using the RNeasy Mini Kit (74104, QIAGEN). The high-capacity cDNA reverse transcription kit (4368814, Life Technologies) was employed to obtain cDNA. Target gene expression (*Cyr61*, *CTGF*, and *ANKRD1*, *Cyp3a*, *Alb*, *Fxr*) was measured with PowerUp SYB Green Master Mix kit (A25777, Life Technologies). $\beta$-actin and *Gapdh* were used as reference genes. The primers are shown as follows:

| *hCyr61* | Forward | **GGAAAAGGCAGCTCACTGAAGC** |
|---|---|---|
| | Reverse | GGAGATACCAGTTCCACAGGTC |
| *hCTGF* | Forward | CTTGCGAAGCTGACCTGGAAGA |
| | Reverse | CCGTCGGTACATACTCCACAGA |
| *hANKRD1* | Forward | CGACTCCTGATTATGTATGGCGC |
| | Reverse | GCTTTGGTTCCATTCTGCCAGTG |
| *h$\beta$-actin* | Forward | CACCATTGGCAATGAGCGGTTC |
| | Reverse | AGGTCTTTGCGGATGTCCACGT |

*Continued on next page*

*Continued*

| | | |
|---|---|---|
| **hCyr61** | **Forward** | **GGAAAAGGCAGCTCACTGAAGC** |
| mCyp3a | Forward | TGGTCAAACGCCTCTCCTTGCTG |
| | Reverse | ACTGGGCCAAAATCCCGCCG |
| mAlb | Forward | GCGCAGATGACAGGGCGGAA |
| | Reverse | GTGCCGTAGCATGCGGGAGG |
| mFxr | Forward | ACAGCTAATGAGGACGACAG |
| | Reverse | GATTTCCTGAGGCATTCTCTG |
| mGapdh | Forward | AGGCCGGTGCTGAGTATGTC |
| | Reverse | TGCCTGCTTCACCACCTTCT |

## RNA-seq analysis

The NCI-H226 cells were treated with vehicle control (DMSO) or TM2, K975, and VT103 at 1 µM for 24 hr. Total RNA was isolated with RNeasy Mini Kit (74104, QIAGEN). The integrity of isolated RNA was analyzed using Bioanalyzer (Agilent Technologies). and the RNA-seq libraries were made by Novogene. All libraries have at least 50 million reads sequenced (150 bp paired-end). The heat-maps were generated using differentially expressed genes from TM2 treatment in NCI-H226 cells with Morpheus (https://software.broadinstitute.org/morpheus/). PCA was determined by PCA function in M3C package in R. GSEA was performed using GSEA software from Broad Institute (http://software.broadinstitute.org/gsea/index.jsp). The YAP_TAZ-TEAD Direct Target Genes sets were generated with the published YAP/TAZ-TEAD target genes (*Zanconato et al., 2015*).

## Cell proliferation assay

NCI-H226, MSTO-211H, H2052, H28, HCT116, and DLD1 cells were seeded at a concentration of 500–2000 cells/well in 100 uL of culture medium in 96-well plates overnight and compounds treated with threefold dilutions of concentrations from 10 µM for 5–7 days. After removal of medium, each well was added 60 µL of MTT reagent (3-(4,5-dimethylthiazol-2-yl)–2,5-diphenyltetrazolium bromide) followed by incubation at 37°C for 4 hr. The absorbance was measured with PerkinElmer EnVision plate reader.

## Drug combination

The drug combination experiments were performed using a drug combination matrix across five doses of TM2 (5 µM, threefold dilution) and nine doses of trametinib (10 µM, threefold dilution) in different tumor cell lines. Cell viability was determined on day 5 after the drugs were administered by MTT. Drug synergy score was calculated following Bliss rule. Synergy score and plot were generated using 'Synergyfinder' package in R language.

## Organoid viability

Mouse hepatic progenitor organoids (70932, STEMCELL Technologies) were seeded in 96-well plate using 20 ul Matrigel (Corning, #354230) and cultured in HepatiCult Organoid Growth Medium (06031, STEMCELL Technologies) with or without TM2. Medium was replaced after every 48 hr with fresh compound. Organoid viability was measured using PrestoBlue HS Cell Viability Reagent (Thermo Fisher, # P50200) following the manufacturer's protocol.

## Organoid size measurement

Mouse hepatic progenitor organoids (70932, STEMCELL Technologies) were treated with vehicle control or TM2 for 48 hr. Size measurement was done in bright-field image using ImageJ software (version 1.52a).

## Immunofluorescence staining

Organoids were plated in 8-well chamber slide and fixed in 4% paraformaldehyde at 4°C for 1 hr. After permeabilization in 0.5% PBST, organoids were blocked with 2% BSA for 2 hr and incubated with primary antibody overnight at 40°C. Imaging was performed on Nikon A1RHD25 confocal microscope.

## Statistics

Data was analyzed using GraphPad Prism 6 and shown as mean ± SEM. All the biochemical experiments are independently repeated at least three times and shown by representative images. Two-tailed $t$-test was used for p-value calculation.

## Synthesis of TEAD inhibitors

All commercially available reagents were used without further purification. All solvents such as ethyl acetate, DMSO, and dichloromethane (DCM) were ordered from Fisher Scientific and Sigma-Aldrich and used as received. Unless otherwise stated, all reactions were conducted under air. Analytical thin-layer chromatography (TLC) plates from Sigma were used to monitor reactions. Flash column chromatography was employed for purification and performed on silica gel (230–400 mesh). $^1$H-NMR were recorded at 500 MHZ on JEOL spectrometer. $^{13}$C NMR were recorded at 125 MHz on JEOL spectrometer. The chemical shifts were determined with residual solvent as internal standard and reported in parts per million (ppm). Compounds S3, S4, S5, S6, TM2 were synthesized through sythetic route shown in *Scheme 1*. Compounds S8 and TM22 were synthesized through sythetic route shown in *Scheme 2*. Compounds S10, S11, S12, S13 and TM22 were synthesized through sythetic route shown in *Scheme 3*. Compounds S15, S16, S17, S18 and TM98 were synthesized through sythetic route shown in *Scheme 4*. Compounds S20, S21 and TM112 were synthesized through sythetic route shown in *Scheme 5*.

**Scheme 1.** Synthetic route for TM2.

## Methyl 3-(2-cyclohexylethoxy)benzoate (S3)

To a solution of methyl 3-hydroxybenzoate S2 (500 mg, 3.29 mmol) in DMF (7 mL) was added (2-bromoethyl)cyclohexane S1 (628.8 mg, 3.29 mmol) and $K_2CO_3$ (628.1 mg, 4.94 mmol). The mixture was then stirred at 110°C for 4 hr. After cooling to temperature, the reaction mixture was diluted with water and extracted with ethyl acetate. The combined organic layer was washed with brine, dried over $Na_2SO_4$, and concentrated in vacuo. The crude residue was purified through silica gel chromatography to give S3 as colorless oil (780 mg, 90%). $^1$H NMR (500 MHz, Chloroform-$d$) $\delta$ 7.61 (d, $J$ = 7.6 Hz, 1H), 7.55 (t, $J$ = 2.1 Hz, 1H), 7.33 (t, $J$ = 7.9 Hz, 1H), 7.09 (dd, $J$ = 8.2, 2.6 Hz, 1H), 4.03 (t, $J$ = 6.7 Hz, 2H), 3.91 (s, 3H), 1.83–1.63 (m, 7H), 1.51 (ttt, $J$ = 10.5, 6.8, 3.5 Hz, 1H), 1.33–1.11 (m, 3H), 0.98 (qd, $J$ = 11.9, 3.3 Hz, 2H).

## 3-(2-Cyclohexylethoxy)benzoic acid (S4)

To a solution of S3 (780 mg, 2.97 mmol) in ethanol (10 mL) was added saturated aqueous KOH (417 µL). The mixture was then stirred at room temperature overnight. After completion, the reaction was quenched with 1 N HCl on ice until PH was adjusted to 1. The mixture was then diluted with water and extracted with ethyl acetate. The combined organic layer was washed with brine, dried

over anhydrous Na$_2$SO$_4$, and concentrated in vacuo to give S4 (650 mg, 88%) which were used directly without further purification.

### *tert*-Butyl 4-(3-(2-cyclohexylethoxy)benzoyl)piperazine-1-carboxylate (S5)

To a solution of S4 (600 mg, 2.42 mmol) in DMF (20 mL) was added HATU (1.38 g, 3.63 mmol) and DIEA (862 µL, 4.84 mmol). After stirring for 5 min, the solution was then added *tert*-butyl piperazine-1-carboxylate (450.6 mg, 2.42 mmol) and continuously stirred at room temperature overnight. After completion, the reaction was quenched with water and extracted with ethyl acetate. The combined organic layer was washed with 1 N HCl, saturated NaHCO$_3$, brine, dried over anhydrous Na$_2$SO$_4$, and concentrated in vacuo. The crude residue was purified through silica gel chromatography to give S5 as a white solid (950 mg, 94%). $^1$H NMR (500 MHz, Chloroform-*d*) $\delta$ 7.30 (t, *J* = 8.0 Hz, 1H), 6.96–6.89 (m, 3H), 3.99 (t, *J* = 6.7 Hz, 2H), 3.82–3.31 (m, 8H), 1.79–1.62 (m, 7H), 1.54–1.39 (m, 1H) 1.47 (s, 9H), 1.32–1.10 (m, 3H), and 0.96 (qd, *J* = 11.9, 3.0 Hz, 2H).

### (3-(2-Cyclohexylethoxy)phenyl)(piperazin-1-yl)methanone (S6)

To a solution of S5 (890 mg, 2.13 mmol) in DCM (4 mL) was added trifluoroacetic acid (4 mL) drop-wise on ice. The mixture was continuously stirred on ice for 30 min. After completion, the reaction was quenched with saturated NaHCO$_3$ dropwise on ice. The mixture was then diluted with water and extracted with ethyl acetate. The combined organic layer was washed with brine, dried over anhydrous Na$_2$SO$_4$, and concentrated in vacuo to give S6, which were used directly without further purification.

### 4-(3-(2-Cyclohexylethoxy)benzoyl)-*N*-phenylpiperazine-1-carboxamide (TM2)

To a solution of S6 (100 mg, 0.403 mmol) in DCM (4 mL) was added isocyanate phenyl isocyanate (63.1 µL, 0.484 mmol). The reaction mixture was stirred at room temperature for 2 hr. The reaction was quenched with water and extracted with DCM. The combined organic layer was washed with brine, dried over anhydrous Na$_2$SO$_4$, and concentrated in vacuo. The crude residue was purified through silica gel chromatography to give TM2 as a white solid (160 mg, 91%). $^1$H NMR (500 MHz, Chloroform-*d*) $\delta$ 7.36–7.24 (m, 5H), 7.04 (t, *J* = 7.3 Hz, 1H), 6.98–6.89 (m, 3H), 6.77 (brs, 1H), 3.99 (t, *J* = 6.7 Hz, 2H), 3.93–3.35 (m, 8H), 1.78–1.62 (m, 7H), 1.54–1.44 (m, 1H), 1.30–1.12 (m, 3H), and 0.97 (qd, *J* = 12.1, 2.9 Hz, 2H). $^{13}$C NMR (125 MHz, Chloroform-*d*) $\delta$ 170.62, 159.45, 155.21, 138.85, 136.47, 129.85, 129.02, 123.55, 120.41, 118.87, 116.46, 113.17, 66.28, 47.46 (brs), 44.22, 42.01 (brs), 36.64, 34.61, 33.39, 26.60, 26.33.

**Scheme 2.** Synthetic route for TM22.

### Phenyl pyridin-3-ylcarbamate (S8)

To a solution of pyridin-3-amine S7 (188.2 mg, 2 mmol) in pyridine (5 mL) was added phenyl chloroformate (274 µL, 2.2 mmol). The reaction mixture *Figure 1* was stirred at room temperature overnight. The mixture was quenched by the addition of ethyl acetate and 10% critic acid. The organic layer was washed with saturated NaHCO$_3$, brine, dried over Na$_2$SO$_4$. The organic solvents were removed in vacuo to give carbamate S8 which was used directly for the next step.

## 4-(3-(2-Cyclohexylethoxy)benzoyl)-*N*-(pyridin-3-yl)piperazine-1-carboxamide (TM22)

To a solution of S6 (30 mg, 0.095 mmol) in DMSO (1 mL) was added carbamate (40.7 mg, 0.19 mmol) and NaOH (114 µL, 0.114 mmol, 10 N). The reaction mixture was stirred at room temperature for 2 hr. The reaction was quenched with water and extracted with ethyl acetate. The combined organic layer was washed with brine, dried over anhydrous $Na_2SO_4$, and concentrated in vacuo. The crude residue was purified through silica gel chromatography to give TM22 as a white solid (36.1 mg, 87%). $^1$H NMR (500 MHz, Chloroform-*d*) δ 8.46 (d, *J* = 2.6 Hz, 1H), 8.26 (dd, *J* = 4.8, 1.4 Hz, 1 H), 7.96 (dt, *J* = 8.4, 2.1 Hz, 1H), 7.34–7.20 (m, 3H), 6.99–6.87 (m, 3H), 3.99 (t, *J* = 6.7 Hz, 2H), 3.88–3.37 (m, 8H), 1.77–1.63 (m, 7H), 1.55–1.45 (m, 1H), 1.29–1.13 (m, 3H), and 0.96 (qd, *J* = 12.0, 2.9 Hz, 2H). $^{13}$C NMR (125 MHz, Chloroform-*d*) δ 170.70, 159.50, 155.05, 144.25, 141.49, 136.36, 136.25, 129.93, 127.78, 123.78, 118.82, 116.49, 113.21, 66.32, 47.43 (brs), 44.24, 42.00 (brs), 36.65, 34.64, 33.41, 26.62, and 26.35.

**Scheme 3.** Synthetic route for TM45.

## Methyl 3-(2-(tetrahydro-2H-pyran-4-yl)ethoxy)benzoate (S10)

To a solution of S9 (400 mg, 3.07 mmol) in anhydrous DCM (20 mL) was added Et$_3$N (642 µL, 4.61 mmol), MsCl (285 µL, 3.68 mmol) at 0°C. The solution was stirred at room temperature. After completion, the reaction mixture was diluted with water, extracted with DCM, and washed with saturated aqueous NaHCO$_3$. The combined organic layer was dried over anhydrous Na$_2$SO$_4$ and concentrated in vacuo to give the methanesulfonate. The methanesulfonate was then dissolved in DMF (10 mL) followed by cautiously adding S2 (513.8 mg, 3.38 mmol) and K$_2$CO$_3$ (848.6 mg, 6.14 mmol). The resulting suspension was further stirred at 80°C for 4 hr. The reaction mixture was extracted with ethyl acetate, then washed with water, brine. The organic phase was dried over anhydrous Na$_2$SO$_4$ and concentrated in vacuo. The crude residue was purified through silica gel chromatography to give S10 as colorless oil (680 mg, 84%). $^1$H NMR (500 MHz, Chloroform-*d*) δ 7.62 (dd, *J* = 7.5, 1.3 Hz, 1H), 7.54 (t, *J* = 2.1 Hz, 1H), 7.33 (t, *J* = 7.9 Hz, 1H), 7.08 (dd, *J* = 8.4, 2.6 Hz, 1H), 4.05 (t, *J* = 6.2 Hz, 2H), 3.97 (ddd, *J* = 11.4, 4.5, 1.7 Hz, 2H), 3.91 (s, 3H), 3.41 (td, *J* = 11.8, 2.1 Hz, 2H), 1.85–1.73 (m, 3H), 1.67 (dq, *J* = 13.3, 2.0 Hz, 2H), and 1.37 (qd, *J* = 11.9, 4.4 Hz, 2H).

## 3-(2-(Tetrahydro-2H-pyran-4-yl)ethoxy)benzoic acid (S11)

S11 was prepared as described for **S4** (670 mg, 2.53 mmol) from **S10** and were used directly without further purification.

## *tert*-Butyl 4-(phenylcarbamoyl)piperazine-1-carboxylate (S12)

S12 was prepared as described for TM2 from *tert*-butyl piperazine-1-carboxylate (1 g, 5.37 mmol) and phenyl isocyanate (767.5 mg, 6.44 mmol) as a white solid (quantitative). $^1$H NMR (500 MHz, Chloroform-*d*) δ 7.35 (d, *J* = 7.6 Hz, 2H), 7.29 (td, *J* = 8.5, 8.0, 2.3 Hz, 2H), 7.08–7.02 (m, 1H), 6.37 (s, 1H), 3.49 (s, 8H), and 1.49 (s, 9H).

## *N*-Phenylpiperazine-1-carboxamide (S13)

S13 was prepared as described for S6 (800 mg, 2.62 mmol) from S12 and were used directly without further purification.

## *N*-Phenyl-4-(3-(2-(tetrahydro-2H-pyran-4-yl)ethoxy)benzoyl)piperazine-1-carboxamide (TM45)

TM45 was prepared as described for S5 from S11 (40 mg, 0.16 mmol) and S13 (39.4 mg, 0.192 mmol) as a white solid (44 mg, 63%). $^1$H NMR (500 MHz, Chloroform-*d*) $\delta$ 7.35–7.29 (m, 3H), 7.29–7.24 (m, 2H), 7.07–7.01 (m, 1H), 6.98–6.89 (m, 3H), 6.73 (s, 1H), 4.01 (t, *J* = 6.1 Hz, 2H), 3.96 (dd, *J* = 11.1, 3.6, 2H), 3.86–3.43 (m, 8H), 3.39 (td, *J* = 11.8, 2.0 Hz, 2H), 1.81–1.71 (m, 3H), 1.68–1.61 (m, 2H), and 1.40–1.30 (m, 2H). $^{13}$C NMR (125 MHz, Chloroform-*d*) $\delta$ 170.52, 159.30, 155.19, 138.85, 136.56, 129.88, 129.01, 123.55, 120.37, 119.03, 116.40, 113.17, 68.06, 65.48, 47.43 (brs), 44.21, 42.01 (brs), 36.15, 33.07, and 32.01.

**Scheme 4.** Synthetic route for TM98.

## *tert*-Butyl 4-(2-(3-(methoxycarbonyl)phenoxy)ethyl)piperidine-1-carboxylate (S15)

S15 was prepared as described for S10 from S14 (480 mg, 2.09 mmol) and S12 (318 mg, 2.09 mmol) as a white solid (530 mg, 70%). $^1$H NMR (500 MHz, Chloroform-*d*) $\delta$ 7.62 (dd, *J* = 7.7, 1.3 Hz, 1H), 7.54 (dd, *J* = 2.7, 1.3 Hz, 1H), 7.33 (t, *J* = 7.9 Hz, 1H), 7.08 (ddd, *J* = 8.3, 2.6, 1.2 Hz, 1H), 4.19–4.05 (m, 2H), 4.05 (t, *J* = 6.1 Hz, 2H), 3.91 (s, 3H), 2.80–2.64 (s, 2H), 1.80–1.65 (m, 5H), 1.46 (s, 9H), and 1.23–1.11 (m, 2H).

## 3-(2-(1-(*tert*-Butoxycarbonyl)piperidin-4-yl)ethoxy)benzoic acid (S16)

S16 was prepared as described for S4 from S15 (380 mg, 1.05 mmol) and were used directly without further purification.

## *tert*-Butyl 4-(2-(3-(4-(phenylcarbamoyl)piperazine-1-carbonyl)phenoxy)ethyl)piperidine-1-carboxylate (S17)

S17 was prepared as described for S5 from S16 (200 mg, 0.572 mmol) and S13 (140.9 mg, 0.686 mmol) as a white solid (270 mg, 88%). $^1$H NMR (500 MHz, Chloroform-*d*) $\delta$ 7.33 (t, *J* = 7.9 Hz, 3H), 7.30–7.24 (m, 2H), 7.03 (tt, *J* = 7.4, 1.3 Hz, 1H), 6.98–6.89 (m, 3H), 6.78 (brs, 1H), 4.16–4.04 (m, 2H), 4.00 (t, *J* = 6.2 Hz, 2H), 3.87–3.35 (m, 8H), 2.70 (s, 2H), 1.82–1.64 (m, 5H), 1.45 (s, 9H), and 1.21–1.11 (m, 2H).

## *N*-Phenyl-4-(3-(2-(piperidin-4-yl)ethoxy)benzoyl)piperazine-1-carboxamide (S18)

S18 was prepared as described for S6 from S17 (175 mg, 0.33 mmol) and were used directly without further purification.

### 4-(3-(2-(1-Acetylpiperidin-4-yl)ethoxy)benzoyl)-*N*-phenylpiperazine-1-carboxamide (TM98)

S18 (25 mg, 0.0573 mmol) was then dissolved in DCM (1.5 mL). The solution was added Et3N (16 μL, 0.115 mmol) and acetyl chloride (4.9 μL, 0.0688 mmol) on ice. The reaction mixture was stirred at room temperature for 2 hr. After completion, the reaction was quenched with saturated $NaHCO_3$ and extracted with ethyl acetate. The combined organic layer was washed with brine, dried over anhydrous $Na_2SO_4$, and concentrated in vacuo. The crude residue was purified through silica gel chromatography to give S18 as a colorless oil (20 mg, 73%). $^1$H NMR (500 MHz, Chloroform-d) $\delta$ 7.38–7.24 (m, 5H), 7.04 (t, *J* = 7.3 Hz, 1H), 6.94 (ddt, *J* = 10.3, 6.1, 2.5 Hz, 3H), 6.86–6.75 (m, 1H), 4.60 (d, *J* = 13.1 Hz, 1H), 4.02 (t, *J* = 5.9 Hz, 2H), 3.92–3.36 (m, 9H), 3.04 (t, *J* = 13.0 Hz, 1H), 2.54 (t, *J* = 13.0 Hz, 1H), 2.08 (s, 3H), 1.84–1.71 (m, 5H), and 1.27–1.10 (m, 2H). $^{13}$C NMR (125 MHz, Chloroform-d) $\delta$ 170.50, 168.98, 159.17, 155.24, 138.92, 136.64, 129.92, 129.02, 123.52, 120.34, 119.17, 116.37, 113.28, 65.57, 46.77, 44.26, 41.90, 35.57, 33.20, 32.78, 31.83, and 21.61. HRMS (ESI): calcd for $C_{27}H_{35}N_4O_4$ [M+H]$^+$, 479.2658; found, 479.2653.

**Scheme 5.** Synthetic route for TM112.

### *N*-(4-Isocyanatophenyl)acetamide (S20)

To a solution of triphosgene (311.6 mg, 1.05 mmol) in DCM (6 mL) was added a solution of Et$_3$N (0.9 mL, 6.45 mmol) and S19 (450.5 mg, 3 mmol) in DCM (6 mL) dropwise on ice. The mixture was continuously stirred at room temperature for 1 hr. The reaction was quenched with saturated $NaHCO_3$ dropwise on ice. The mixture was then diluted with water and extracted with ethyl acetate. The combined organic layer was washed with brine, dried over anhydrous $Na_2SO_4$, and concentrated in vacuo to give S20 which were used directly without further purification.

### *N*-(4-Acetamidophenyl)-4-(3-(2-cyclohexylethoxy)benzoyl)piperazine-1-carboxamide (S21)

S21 was prepared as described for TM2 from S6 (120 mg, 0.376 mmol) and *N*-(3-isocyanatophenyl)acetamide (79.5 mg, 0.451 mmol) as a white solid (100.5 mg, 54%). $^1$H NMR (500 MHz, Chloroform-d) $\delta$ 7.40 (d, *J* = 8.4 Hz, 2H), 7.33–7.26 (m, 3H), 7.15 (brs, 1H), 6.98–6.89 (m, 3H), 6.39 (brs, 1H), 3.99 (t, *J* = 6.7 Hz, 3H), 3.92–3.35 (m, 8H), 2.15 (s, 3H), 1.77–1.62 (m, 7H), 1.53–1.44 (m, 1H), 1.29–1.10 (m, 3H), and 1.01–0.90 (m, 2H).

### *N*-(4-Aminophenyl)-4-(3-(2-cyclohexylethoxy)benzoyl)piperazine-1-carboxamide (TM112)

To a solution of S21 (80 mg, 0.161 mmol) in methanol (2 mL) was added 2 N HCl (4 mL). The reaction was refluxed for 2 hr. After cooling down to room temperature, the reaction mixture was basified with saturated $NaHCO_3$ on ice and extracted with ethyl acetate. The combined organic layer was washed with brine, dried over anhydrous $Na_2SO_4$, and concentrated in vacuo. The crude residue was purified through silica gel chromatography to give TM112 as colorless oil (23.8 mg, 33%). $^1$H NMR (500 MHz, Chloroform-d) $\delta$ 7.31 (t, *J* = 8.0 Hz, 1H), 7.09 (d, *J* = 8.6 Hz, 2H), 6.98–6.90 (m, 3H), 6.63 (d, *J* = 8.6 Hz, 2H), 6.23 (s, 1H), 4.00 (t, *J* = 6.7 Hz, 2H), 3.93–3.26 (m, 10H), 1.78–1.64 (m, 7H), 1.55–1.45 (m, 1H),

1.31–1.15 (m, 3H), and 0.97 (qd, $J$ = 12.1, 3.0 Hz, 2H). $^{13}$C NMR (125 MHz, Chloroform-$d$) $\delta$ 170.63, 159.47, 155.85, 143.21, 136.61, 129.85, 129.80, 123.29, 118.97, 116.49, 115.70, 113.21, 66.32, 44.26, 36.68, 34.66, 33.42, 26.64, and 26.37.

## Acknowledgements

The high-throughput screen to identify TM2 series of compounds was supported by a sponsored research agreement with Astellas Innovation fund and carried out with Astellas non-proprietary compound collections. We thank NIH fundings (R01CA219814 and R01CA238270 to XW, R01DK127180 and R01DK127207 to JM) and grants from the Welch Foundation (I-1932 to XL). HL is partly supported by a postdoc fellowship from Antidote Health Foundation for Cure of Cancer. We thank Ms. Qian Xu for technical assistance with cloning and RT-PCR experiments.

## Additional information

### Competing interests

Lu Hu: is an inventor of a patent application covering TM2 and analogues as novel TEAD inhibitors. Xu Wu: is an inventor of a patent application covering TM2 and analogues as novel TEAD inhibitors. Dr. Xu Wu has a financial interest in Tasca Therapeutics, which is developing small molecule modulators of TEAD palmitoylation and transcription factors. Dr. Wu's interests were reviewed and are managed by Mass General Hospital, and Mass General Brigham in accordance with their conflict of interest policies. The other authors declare that no competing interests exist.

### Funding

| Funder | Grant reference number | Author |
|---|---|---|
| National Cancer Institute | R01CA219814 | Xu Wu |
| National Cancer Institute | R01CA238270 | Xu Wu |
| National Institute of Diabetes and Digestive and Kidney Diseases | R01DK127180 | Junhao Mao |
| National Institute of Diabetes and Digestive and Kidney Diseases | R01DK127207 | Junhao Mao |
| Welch Foundation | I-1932 | Xuelian Luo |
| Antidote Health Foundation for the cure of cancer | postdoc fellowship | Lu Hu |

The funders had no role in study design, data collection and interpretation, or the decision to submit the work for publication.

### Author contributions

Lu Hu, Conceptualization, Data curation, Formal analysis, Investigation, Writing – original draft, Writing – review and editing; Yang Sun, Data curation, Formal analysis, Investigation, Writing – review and editing; Shun Liu, Data curation, Formal analysis, Investigation; Hannah Erb, Data curation, Investigation, Writing – review and editing; Alka Singh, Data curation, Formal analysis; Junhao Mao, Formal analysis, Funding acquisition, Writing – review and editing; Xuelian Luo, Formal analysis, Funding acquisition, Investigation, Writing – review and editing; Xu Wu, Conceptualization, Formal analysis, Supervision, Funding acquisition, Writing – original draft, Writing – review and editing

### Author ORCIDs

Lu Hu http://orcid.org/0000-0002-1594-8828
Shun Liu http://orcid.org/0000-0002-1766-2057
Junhao Mao http://orcid.org/0000-0003-1980-1177
Xuelian Luo http://orcid.org/0000-0002-5058-4695

Xu Wu http://orcid.org/0000-0002-1624-0143

**Decision letter and Author response**
Decision letter https://doi.org/10.7554/eLife.80210.sa1
Author response https://doi.org/10.7554/eLife.80210.sa2

## Additional files

### Supplementary files
• MDAR checklist

### Data availability
The crystal structure of TEAD2 YBD in complex with TM2 has been deposited in the Protein Data Bank with accession codes 8CUH. The raw RNA-seq data of NCI-H226 treated with TM2, K975 and VT103 has been deposited in NCBI GEO and is accessible at https://www.ncbi.nlm.nih.gov/geo/query/acc.cgi?acc=GSE215114.

The following datasets were generated:

| Author(s) | Year | Dataset title | Dataset URL | Database and Identifier |
|---|---|---|---|---|
| Wu X, Sun Y, Hu L | 2022 | Next Generation Sequencing Quantitative Analysis of TEAD Inhibitors in NCI-H226 cells | https://www.ncbi.nlm.nih.gov/geo/query/acc.cgi?acc=GSE215114 | NCBI Gene Expression Omnibus, GSE215114 |
| Hu L, Sun Y, Liu S, Erb H, Singh A, Mao J, Luo X, Wu X | 2022 | Discovery of a new class of reversible TEA-domain Transcription factors inhibitors with a novel binding mode | https://doi.org/10.2210/pdb8CUH/pdb | Worldwide Protein Data Bank, 10.2210/pdb8CUH/pdb |

The following previously published dataset was used:

| Author(s) | Year | Dataset title | Dataset URL | Database and Identifier |
|---|---|---|---|---|
| Calvet L, Dos-Santos O, Spanakis E, Valence S, Jean-Baptiste V, Le Bail J, Buzy A, Paul P, Henry C, Pollard J, Sidhu S, Moll J, Debussche L, Valtingojer I | 2022 | Human malignant mesothelioma NCI-H226 cells treated with TEAD inhibitor K-975 in SCID mice | https://www.ncbi.nlm.nih.gov/geo/query/acc.cgi?acc=GSE196726 | NCBI Gene Expression Omnibus, GSE196726 |

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
