## [Editor Report]

In this article, Hu et al. describe the discovery and characterization of a new class of reversible TEAD inhibitors that binds to a novel side pocket adjacent to the palmitate-binding pocket. The newly identified highly tractable chemical matter and its novel binding mode provide an excellent starting point for the development of effective TEAD inhibitors.

---

## [Decision Letter]

**Decision letter after peer review:**

Thank you for submitting your article "Discovery of a new class of reversible TEA-domain transcription factor inhibitors with a novel binding mode" for consideration by *eLife*. Your article has been reviewed by 2 peer reviewers, and the evaluation has been overseen by a Reviewing Editor and Kevin Struhl as the Senior Editor. The reviewers have opted to remain anonymous.

Essential revisions:

In this manuscript, Hu et al., describe the discovery and characterization of a new class of reversible TEAD inhibitors that binds to a novel side pocket adjacent to the palmitate-binding pocket. While reviewers recognize the potential significance of the new compound and its novel binding mode, they also expressed some concerns about whether the study represents a significant enough advance for *eLife*. They suggest the following essential revisions:

1) The authors need to demonstrate that occupation of the new pocket indeed contributes to improved potency and/or specificity over known inhibitors. In this regard, it is important that a direct comparison of the efficacy between TM2 and known inhibitors should be performed in the same cell type under similar conditions.

2) Statistical analysis is generally lacking. Quantitation and a description of how many times experiments were performed should be included throughout. Especially, RNA-seq should be performed with a minimum of triplicates, and the authors need to compare an RNA-seq dataset involving the known inhibitor using an identical cell-based system (time point, cell line).

*Reviewer #1 (Recommendations for the authors):*

1. The claim of specificity based on a comparison of RNA-Seq datasets derived from an in vivo experiment versus a cell-based monoculture assay is not relevant. Direct comparison to K975 can only be made using the same time point/duration of target engagement and cell type.

2. It would substantially help to clarify any differences (or identity) in the side chains within the newly identified inhibitor binding site across TEAD family members, to substantiate the claim that this is a new site. It appears to be the case, but the inhibitors used for comparison were structurally characterized in the context of different (not TEAD2) family members (if I understand correctly).

3. The synergy data is not overly convincing. How many well replicates were used to generate the number that is shown? Can statistical significance with respect to the difference between wells under a given condition be given?

*Reviewer #2 (Recommendations for the authors):*

– The identification of new regions within the TEAD palmitate-binding pocket that may impact TEAD functions is interesting, but it remains unclear if/how these residues influence YAP binding or TEAD activity beyond other inhibitors that sit in the hydrophobic pocket.

– Given the statistical limitations, it is unclear why the authors chose to perform RNA-seq on two replicates to characterize the effects of TM2 treatment on gene expression. It is also unclear how the authors defined TM2-regulated genes based only on replicates. A minimum of triplicate samples for such analysis is suggested.

– It is unclear what the analysis in Figure 3 – supplemental 2 is trying to show. If the goal is to show that similar genes are affected by both K975 and TM2 then a global analysis, rather than focusing on previously reported YAP/TAZ targets, would be more appropriate. If the goal is to state that TM2 is more potent, then analysis under the same condition needs to be performed. The comparison of in vivo animal treatment versus in vitro cell culture treatment is not informative.

– The effects of TM2 treatment on the organoids could benefit from quantitation across multiple experiments, particularly the effects on proliferation. It would also be important to quantify the organoid size and map the effects of TM2 treatment on the fate of the organoids (i.e., does TM2 treatment promote cell differentiation?).

– Quantitation and a description of how many times experiments were performed should be included throughout.

---

## [Author Response]

Essential revisions:In this manuscript, Hu et al., describe the discovery and characterization of a new class of reversible TEAD inhibitors that binds to a novel side pocket adjacent to the palmitate-binding pocket. While reviewers recognize the potential significance of the new compound and its novel binding mode, they also expressed some concerns about whether the study represents a significant enough advance for eLife. They suggest the following essential revisions:1) The authors need to demonstrate that occupation of the new pocket indeed contributes to improved potency and/or specificity over known inhibitors. In this regard, it is important that a direct comparison of the efficacy between TM2 and known inhibitors should be performed in the same cell type under similar conditions.

We appreciate the reviewers’ suggestion to directly compare the efficacy of TM2 and other known inhibitors. In the revised manuscript, we compared TM2 with K975 (Kirin, irreversible inhibitor), VT103 (Vivace, reversible inhibitor), and our earlier published TEAD inhibitor MGH-CP1 side-by-side. First, in TEAD2 auto-palmitoylation assay, our results showed that TM2 is more potent than K975, VT103 and MGH-CP1 (Figure 1—figure supplement 2A). Similar results were also observed when we compared their potencies on TEAD4 auto-palmitoylation (Figure 1—figure supplement 2B). Next, we evaluated their activities to inhibit full length myc-TEAD1 palmitoylation in cells. TM2, K975 and VT103, significantly suppressed Myc-TEAD1 palmitoylation under submicromolar concentration (Figure 1—figure supplement 2C). Among them, VT103 is slightly more potent on TEAD1. This is consistent with the published report that VT103 is a TEAD1 specific inhibitor, but with less activities on TEAD2 and TEAD4 (Tang *et al., 2021*, *Mol. Cancer Ther.*). These results suggested that TM2 showed broader pan-inhibitory effects on palmitoylation of TEAD family. Consistently, TM2 showed the strongest inhibition on endogenous pan-TEAD palmitoylation (Figure 1—figure supplement 2D). Taken together, these new data suggested that TM2 is a more potent and broader pan-inhibitor of TEAD family compared to known inhibitors. We have added these data to new Figure 1—figure supplement 2.

We then compared the antiproliferation effects of TM2, K975, and VT103 in YAP-dependent mesothelioma cells. In NCI-H226 cells (NF2 mutant), TM2 showed potent and comparable activity with K975, an irreversible inhibitor. Surprisingly, VT103 showed a strikingly low nM IC50 value, which is even lower than its on-target inhibition on TEAD1 (Figure 4—figure supplement 3A), suggesting that its anti-proliferative activities might be due to potential off-target effects. Furthermore, we also tested these compounds in MSTO-211H cells with strong YAP dependency due to Lats1/2 deletion. Consistent with the observation in NCI-H226, the covalent inhibitor K975 showed comparable potency to TM2. However, non-covalent inhibitor VT103 did not exhibit meaningful inhibition and showed a flat curve (data not shown). Although VT103 suppressed cell growth to about 40% under low nanomolar concentrations as MTT assay indicated, we did not observe obvious cell growth inhibition under microscope when counting cell numbers (data not shown), suggesting that VT103’s effects might be due to non-specific inhibition of MTT conversion. The results are consistent with previous paper showing that VT103 did not inhibit proliferation of MSTO-211H cells (Tang et al., 2021, PMID: 33850002). Overall, TM2 and K975 displayed best inhibitory effects on cell proliferation in mesothelioma cells. Given that irreversible inhibitors harbor off-target disadvantages, and K975 has reported to have kidney toxicity (Kaneda *et al., Am. J. Cancer Res.* 2020, PMID: 33415007), a more potent reversible inhibitor, such as TM2, could be more promising therapeutic agent. This data has been added to new Figure 4—figure supplement 3.

2) Statistical analysis is generally lacking. Quantitation and a description of how many times experiments were performed should be included throughout. Especially, RNA-seq should be performed with a minimum of triplicates, and the authors need to compare an RNA-seq dataset involving the known inhibitor using an identical cell-based system (time point, cell line).

We thank the reviewers for the thoughtful comments. In the revised manuscript, we included the description that indicates how many replicates were performed for every quantification experiment in the Figure legend. As suggested, we have performed RNA-seq analyses with 3 independent replicates for each sample. As Figure 3C-F shown, TM2 showed significant inhibition on transcriptional activities of Hippo pathway. Compared with two other TEAD inhibitor, K975 and VT103, TM2 showed a similar pattern on regulation of global gene expressions (Figure 3—figure supplement 2). To further specify their regulation on Hippo transcriptional outputs, we compared the alterations of YAP/TAZ-TEAD target genes in NCI-H266 treated with these three inhibitors individually with the same time point and concentrations. TM2 displayed comparable potency with the irreversible inhibitor K975 on target genes regulation visualized by heatmap analysis (Figure 3G), and both compounds showed stronger suppression than VT103. Gene Set Enrichment Analysis (GSEA) of TEAD-YAP/TAZ target genes for TM2, K975 and VT103 was also performed with the normalized enrichment score (NES) of -2.64, -2.56, -2.31, respectively (Figure 3—figure supplement 3), suggesting that TM2 likely showed the best TEAD-YAP target gene enrichment (more negative value means stronger suppression of the gene set).

Reviewer #1 (Recommendations for the authors):1. The claim of specificity based on a comparison of RNA-Seq datasets derived from an in vivo experiment versus a cell-based monoculture assay is not relevant. Direct comparison to K975 can only be made using the same time point/duration of target engagement and cell type.

We agree reviewer 1’s comments. We generated new triplicated RNA-seq datasets using H226 cells with comparison among TM2, K975 and VT103. New panels are shown in Figure 3 and Figure 3—figure supplement 2-3.

2. It would substantially help to clarify any differences (or identity) in the side chains within the newly identified inhibitor binding site across TEAD family members, to substantiate the claim that this is a new site. It appears to be the case, but the inhibitors used for comparison were structurally characterized in the context of different (not TEAD2) family members (if I understand correctly).

We appreciate the reviewer 1’s insightful suggestion. We here aligned crystal structures of TEAD1-4 and found all eight interacting residues in the new side pocket of TEAD2 were highly conserved among all the TEAD family members (Figure 2—figure supplement 2A). In addition, the protein sequence alignment also demonstrated that the key residues with the newly identified binding site were highly conserved, even in TEADs from other species (Figure 2—figure supplement 2B). Although there are some variants, for example, Cys343 is a Val in TEAD3 and Q410 is Leu in TEAD1, the binding affinity should not be affected.

3. The synergy data is not overly convincing. How many well replicates were used to generate the number that is shown? Can statistical significance with respect to the difference between wells under a given condition be given?

We have updated the figure legend to illustrate what the panels showed in Fig4E. Figure 4E Left panels show the dose curve of the single agents. Middle panel showed the heatmap with color-coding as percentage of cell viability normalized to untreated controls under combination. Right panel shows the heatmap of Bliss score for TM2 and Trametinib combination. The Synergy score was generated by “Synergyfinder” package in R software (PMID: 35580060). Bliss score of higher than 10 indicates synergistic effects. In Fig4E, we observed strong synergy (Bliss score 20-30) across multiple doses. This is a commonly used method to determine the combinational effect using heatmaps based on Bliss score, in which the statistical models are not generally carried out (PMID: 34234012; PMID: 32368721; PMID: 27165605).

Reviewer #2 (Recommendations for the authors):– The identification of new regions within the TEAD palmitate-binding pocket that may impact TEAD functions is interesting, but it remains unclear if/how these residues influence YAP binding or TEAD activity beyond other inhibitors that sit in the hydrophobic pocket.

We did not observe significant conformational changes in TEAD-YAP binding interface in TEAD2 YAP binding domain (YBD)-TM2 co-crystal structures. The inhibition of TEAD-YAP binding by TEAD inhibitors (including all TEAD inhibitor bound to the lipid binding site) has been observed in cell based assay with full length TEAD proteins, suggesting that additional factor or mechanisms are involved, and full-length TEADs might be required. This is consistent with other published TEAD inhibitors, including K975 and VT103, which do not significantly alter the conformation of TEAD YBD. The binding to the additional pocket by TM2 might contribute to enhanced affinity of TM2 to TEADs and its pan-TEAD inhibition. It is possible that TM2 might induce changes in full-length TEADs structures, which remain difficult to be determined by X-ray crystallography. However, we have observed that TM2 could disrupt TEAD-TEAD dimerization (data not shown), and more studies are needed to understand the detailed mechanisms and functional consequences. We would publish these data in future publications.

– Given the statistical limitations, it is unclear why the authors chose to perform RNA-seq on two replicates to characterize the effects of TM2 treatment on gene expression. It is also unclear how the authors defined TM2-regulated genes based only on replicates. A minimum of triplicate samples for such analysis is suggested.

We thank the reviewer for this comment. As suggested, we have performed RNA-seq analysis with 3 independent replicates for each sample. We have added these data in Figure 3 and Figure 3—figure supplement 2-3.

– It is unclear what the analysis in Figure 3 – supplemental 2 is trying to show. If the goal is to show that similar genes are affected by both K975 and TM2 then a global analysis, rather than focusing on previously reported YAP/TAZ targets, would be more appropriate. If the goal is to state that TM2 is more potent, then analysis under the same condition needs to be performed. The comparison of in vivo animal treatment versus in vitro cell culture treatment is not informative.

We thank reviewer 2’s comments. We agree that the comparison of in vivo animal treatment versus in vitro cell culture treatment is not informative. We generated new triplicated RNA-seq datasets using NCI-H226 treated with vehicle control, TM2, K975 and VT103. The new panel with comparison of target genes are shown in Figure 3G and Figure 3—figure supplement 2-3.

– The effects of TM2 treatment on the organoids could benefit from quantitation across multiple experiments, particularly the effects on proliferation. It would also be important to quantify the organoid size and map the effects of TM2 treatment on the fate of the organoids (i.e., does TM2 treatment promote cell differentiation?).

We appreciate reviewer 2’s insightful comments. As suggested, Ki67 staining was quantified across biological independent replicates (n = 3) for each group, which confirmed that TM2 disrupted organoid maintenance (New Figure 4B). To study the effects of TM2 on organoid growth, we measured the size of hepatic organoids. Our results showed that organoid size was significantly inhibited when treated with TM2 (Figure 4—figure supplement 2A). Recent evidence demonstrated that activation of Hippo signaling is required to maintain differentiation state of hepatocyte (PMID: 24906150; PMID: 27358050). Therefore, we investigated if inhibition of TEAD-YAP activity will affect the fate of organoids by detecting expression of mature hepatocyte markers, such as *Alb*, *Fxr* and *cyp3a*. Although TM2 only showed minor effect on *cyp3a*, it significantly induced expressions of *Alb* and *Fxr* (Figure 4—figure supplement 2B)*,* suggesting TM2 might induce hepatic differentiation. These data were added in the new Figure 4B and Figure 4—figure supplement 2.

– Quantitation and a description of how many times experiments were performed should be included throughout.

We thank the reviewer for pointing out this issue. In the revised manuscript, we included the description that indicates how many replicates were performed for every experiment with quantification in the Figure legend.